

# Validation of improved cytochrome c oxidase I (COI) primers for comprehensive biodiversity assessment of ascidians

Seongjun Bae

National Marine Biodiversity Institue of Korea, Seocheon, Republic of Korea

## ABSTRACT

Reliable molecular tools are needed for effective biodiversity assessment of marine organisms. These tools can be used in ascidians species, which are one of the most invasive taxa worldwide and morphologically difficult to identify. This study aimed to redesign and improve ascidian-specific primers and validate the new primer pair (AscCOI2) for comprehensive biodiversity assessment. To design an optimized primer, 3,948 COI sequences from 273 ascidian species were used as a dataset. The AscCOI2 pair was developed through strategic modifications to the binding site and validated using *in silico* and *in vitro* approaches. Analysis of penalty scores showed the improved efficiency of the redesigned AscCOI2 pair, with ascidians scoring less than 150 points for both forward and reverse primers and the non-target groups maintaining a score above 480. Primer-binding analysis results showed a significant improvement in amplification success rate from 47.99% to 82.42% at the species level. *In vitro* validation using conventional polymerase chain reaction (PCR) confirmed the successful amplification of six ascidian species and failure for a non- ascidian taxon. Barcoding gap analysis showed a clear gap of 0.015 between intraspecific and interspecific genetic distances. The species detection capability of the redesigned AscCOI2 pair greatly improved, and the high taxonomic specificity of ascidians is maintained. Overall, this study demonstrates that the AscCOI2 pair is an effective tool for both metabarcoding and mini-barcode applications in biodiversity assessment and molecular systematics research of ascidians.

# INTRODUCTION

Understanding and monitoring of ecosystem biodiversity are fundamental for effective environmental management and conservation (*Campbell et al., 2002*). Traditional biodiversity assessments have heavily relied on visual surveys for morphological identification and are often constrained by time, resource, and expertise (*Watts et al., 2019*; *Gold et al., 2021*).

To overcome these limitations, DNA barcoding emerged as a relatively simple and efficient method of species identification. This method uses short, standardized genetic markers to identify and differentiate species across a wide range of taxa. The mitochondrial cytochrome c oxidase I (COI) gene serves as the main barcode marker for animals, whereas internal transcribed spacer (ITS) and 16S are used for fungi and bacteria, respectively

Corresponding author
Seongjun Bae, silverto@naver.com

(*Stackebrandt & Goebel, 1994*; *Hebert et al., 2003a*; *Schoch et al., 2012*). The effectiveness of DNA barcoding depends on the "barcoding gap", the clear separation between intraspecific and interspecific genetic variation that allows reliable species identification (*Meyer & Paulay, 2005*). Accurate species identification is possible if the interspecific genetic variation is greater than the intraspecific genetic variation and there is sufficient space (barcoding gap) between the two variations (intraspecific and interspecific).

Environmental DNA (eDNA) metabarcoding analysis provides a comprehensive approach and has become a powerful complementary tool for monitoring biodiversity in marine ecosystems, owing to its spatial and temporal efficiency and application in a wide range of taxa (*Capurso, Carroll & Stewart, 2023*). This method is especially useful for marine biofouling organisms such as ascidians, many of which are potentially invasive and difficult to identify morphologically (*Bae, Kim & Yi, 2023*). Additionally, biodiversity assessment is particularly difficult in marine ecosystems, in which DNA degrades relatively fast (*Collins et al., 2018*; *Wood et al., 2020*). For this reason, the effectiveness of the DNA metabarcoding tools cannot be successfully used unless they are designed and selected appropriately. Although studies on high-throughput sequencing technology have facilitated biodiversity monitoring in these environments (*Garlapati et al., 2019*), the advances of such technology critically depend on primer design and selection (*Valentini et al., 2016*). Mini-barcode primers that target fragmented DNA have proven to be effective in marine eDNA metabarcoding studies as they improve amplification success and efficiency (*Leray et al., 2013*).

Therefore, new specific primers have been developed for various taxonomic groups to achieve short amplicon lengths (*Sbrana et al., 2024*; *Zimmermann, Har ardottir & Ribeiro, 2024*). In the case of marine lobster, the development of a 230 bp mini-barcode primer has been shown to improve species identification compared to universal primers (*Govender et al., 2019*). *Leray et al. (2013)* designed a primer pair that targets a 313 bp COI fragment, and it proved to be effective for metazoan diversity.

Previous research has demonstrated that ascidian-specific primers targeting mini-barcode regions can successfully detect a wide range of ascidian species in seawater samples (*Bae, Kim & Yi, 2023*). However, some species (*e.g.*, *Didemnum vexillum*) in South Korea (*Bae et al., 2023*) have not been detected by these primers, suggesting the need to improve the ascidian-specific primer pair to enlarge its spectrum of species detection. Moreover, primer-template mismatches are known to introduce significant bias in species detection (*Piñol et al., 2015*), emphasizing the need for improved primers that combine both high amplification efficiency and strong taxonomic specificity.

To address these challenges, this study aimed to generate a new ascidian-specific primer pair (AscCOI2) by improving the previous pair AscCOI (*Bae, Kim & Yi, 2023*). To ensure the quality of the new pair, this study aimed to (1) verify the primer-template binding efficiency and ascidian specificity through *in silico* analysis and (2) confirm the performance of the new primer pair using *in vitro* analysis and conventional polymerase chain reaction (PCR). Results of this cross-validation demonstrate that AscCOI2 is an effective tool for both mini-barcode and eDNA metabarcoding applications in ascidian biodiversity assessment.

## MATERIALS AND METHODS

### Improvement and optimization of primers

COI sequences for dataset construction of target group (ascidian; Supplementary Information 1) and non-target groups (Anthozoa, Bivalvia, Gymnolaemata, Hydrozoa, Porifera, Thecostraca, Echinodermata, and Chondrichthyes; Supplementary Information 2) were obtained from the National Center for Biotechnology Information (NCBI) GenBank (https://www.ncbi.nlm.nih.gov, accessed June 2024). For ascidians, we collected all available COI sequences, including those from complete mitochondrial genomes. For non-target groups, we collected individual partial COI sequences. The final sequence dataset consisted of 3,948 ascidian COI sequences belonging to 238 species, 46 genera, and 16 families, and 78,431 COI sequences of non-target groups (benthic invertebrates and cartilaginous fishes), comprising eight taxonomic groups: Anthozoa (8,449 sequences), Bivalvia (2,917 sequences), Gymnolaemata (1,810 sequences), Hydrozoa (4,461 sequences), Porifera (6,594 sequences), Thecostraca (14,122 sequences), Echinodermata (19,905), and Chondrichthyes (20,173).

The sequences were aligned using Clustal Omega in Geneious Prime 2024.0.5 (Biomatters Ltd., Auckland, New Zealand). Primer binding regions were strategically redesigned to improve species detection coverage and maintain ascidian-specificity. The new forward primer (hereinafter AscCOI2_F) only changed the sequence from the forward primer of the previous study (hereinafter AscCOI_F; *Bae, Kim & Yi, 2023*). For the new reverse primer (hereinafter AscCOI2_R), we strategically redesigned it to detect more conserved regions within the COI gene by relocating the binding site by 21 bp in the 3′ direction from the reverse primer of the previous study (hereinafter AscCOI_R). This optimization resulted in an amplicon (216 bp) that was slightly longer than that of the AscCOI_R (197 bp; Table 1 and Fig. 1B).

The preservation areas suitable for primer binding were confirmed through similarity index analysis. The sequence similarity index was calculated using 1/D, where D represents the 'degree of degeneration' of each potential primer sequence (*Hoareau & Boissin, 2010*). In this study, the 'degree of degeneration' refers to the number of possible sequence variants at a given position due to ambiguous bases, and 'similarity' refers to the level of conservation of all sequences at each potential binding site. The higher the similarity index, the lower the sequence variation and higher the conservation. The value of the similarity index was calculated using a sliding window approach (window size: 20 bp, step size: one bp), based on the consensus sequence derived from the aligned sequence dataset, to ensure consistent positioning. In addition, the reliability of the conserved region was confirmed by calculating the proportion of sequences present at each position and displaying it alongside the similarity index. The similarity index was calculated using custom functions based on the 'Biostrings' package (*Pagès et al., 2024*) in R version 4.0.2 (*R Core Team, 2021*). A custom function was implemented to compute 1/D, where D represents the product of degeneracy values for each base in the sequence window.

**Table 1 Details information of ascidian-specific primers.** Details of ascidian-specific primers designed in this study and previous study.

| Primer label | Sequence (5′–3′) | Tm (°C) | Amplicon size (bp) | Target gene | Reference |
|---|---|---|---|---|---|
| AscCOI_F | CCTGATATGGCNTTYCCHCG | 57.3 | 197 | Mitochondrial COI | *Bae et al. (2022)* |
| AscCOI_R | GCTAAATGHAAHGAAAAAATWGC | 51.7 | | | |
| AscCOI2_F | CCDGATATAGCHTTYCCWCG | 52.9 | 216 | Mitochondrial COI | This study |
| AscCOI2_R | CYTAAAATACTWGAAACNCTHGC | 50.3 | | | |

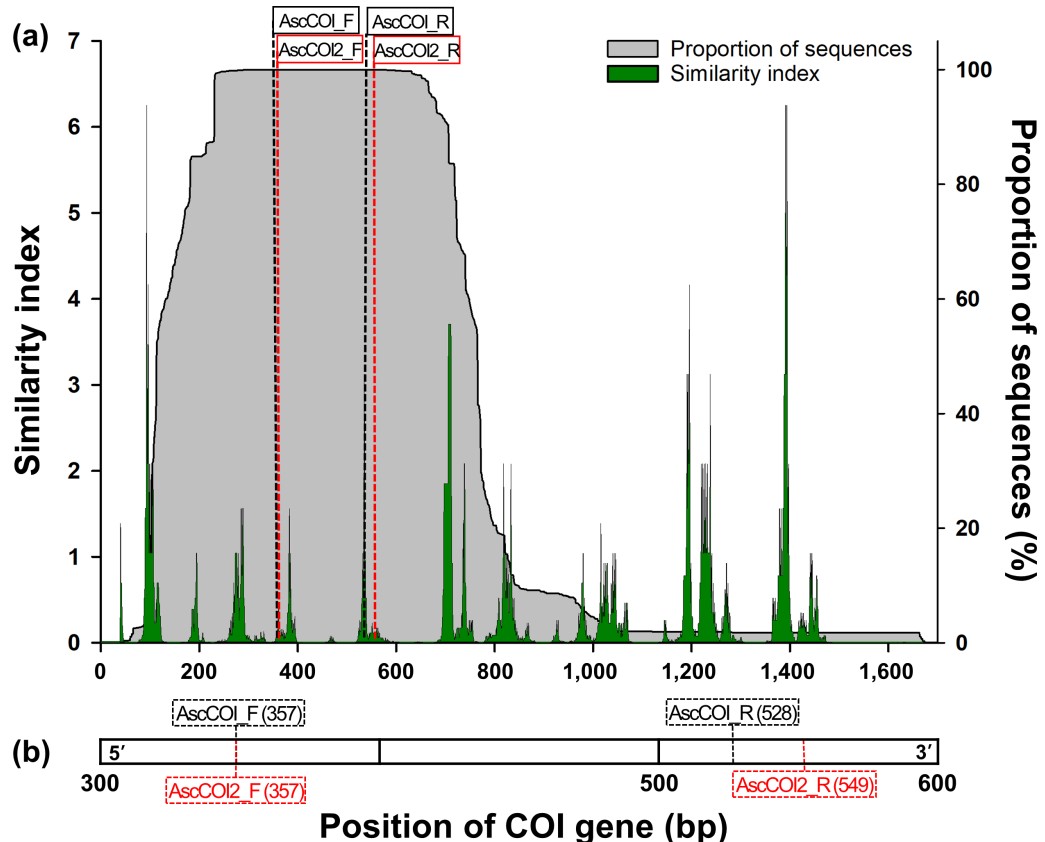

**Figure 1** (A) Similarity index and sequence proportion distribution in the COI gene region. The green histogram/areas represent the sequence similarity index, and the shaded grey histogram/areas represent the proportion of sequences. The black- and red-dashed vertical lines indicate the positions of the AscCOI and AscCOI2 primer pairs, respectively. (B) Detailed location indication of AscCOI_F (357), AscCOI_R (528), AscCOI2_F (357), and AscCOI2_R (549). The numbers in parentheses indicate the starting position based on 5′–3′.

## *In silico* validation

*In silico* analysis was used to evaluate primer binding efficiency, and primer pair performance was assessed using penalty scores for both target (ascidians) and non-target (benthic invertebrates and cartilaginous fishes) groups. The binding analysis was performed using Geneious Prime to allow for a maximum of three bp mismatch in the binding region and two bp mismatch at the 3′ end (*Christopherson, Sninsky & Kwok, 1997*; *Miura et al.,*

*2005*). For penalty score analysis, we used the 'PrimerMiner' package (*Elbrecht & Leese, 2017*) using the 'evaluate_primer' function with the following parameters: mm_position = 'Position_v1', mm_type = 'Type_v1', adjacent = 1, gap_NA = FALSE, and N_NA = FALSE. The 'Position_v1' score matrix is for penalties by position (assigning a higher penalty to end-of-3 mismatches) and the 'Type_v1' matrix is for mismatch type penalties. No additional penalty was applied for adjacent mismatches, and the gaps and 'N's were treated as missing data and assigned penalty scores. Considering the location and type of mismatches between primers and template sequences, mismatches at the 3′ end of primers were given a higher penalty according to the Position_v1 matrix. The penalty for adjacent mismatches increased and then decreased when the wobble base of the target sequence partially matched the primers. Penalty score analysis was performed using both AscCOI and AscCOI2 for target groups and only AscCOI2 for non-target groups.

The performance of primer pairs was evaluated using a combined penalty score approach. For each sequence, penalty scores from forward and reverse primers were summed, and a total threshold of 150 was used to indicate the amplification success. The threshold was set at 150, which is the median of the mid-range (100–200; *Bylemans et al., 2018*), and scores below this threshold indicate acceptable primer binding efficiency. This mid-range represents an empirically established penalty score zone in which primer performance transitions from amplification success (below 100) to amplification failure (above 200). Sequences with combined scores below 150 indicate likely successful amplification and were classified as "Working." Those that exceeded the threshold suggest probable amplification failure and were classified as "Failed." Sequences with incomplete data, which prevented full evaluation, were marked as "Missing data/gaps." This evaluation was performed for both target and non-target groups to assess both primer efficiency and specificity.

To statistically compare the amplification efficiency of the AscCOI and AscCOI2, a generalized linear mixed model (GLMM) analysis was performed using the 'lme4' package (*Bates et al., 2014*). This model explains the differences in amplification success between species by setting amplification success (binary: working/failed) as the response variable, primer type (AscCOI *vs.* AscCOI2) as the fixed effect, and species as the random effect. Statistical significance was assessed using the Wald z-test, and the effect size was calculated as the odds ratio of the model coefficient. To evaluate the species identification capability of AscCOI2, a barcoding gap analysis was performed. The genetic distance was calculated with only the target ascidian sequences using the K80 model implemented in the 'barcoding.gap' function of the 'BarcodingR' package (*Zhang et al., 2017*).

### *In vitro* validation

For *in vitro* validation, primers were tested using conventional PCR to target the genomic DNA (gDNA) of six target species and 16 non-target species (Table S1). Genomic DNA was extracted using the DNeasy Blood and Tissue Kit (QIAGEN, Hilden, Germany) and amplified using AccuPower PCR PreMix (BIONEER, Daejeon, South Korea). Each PCR was carried out in an AccuPower PCR PreMix tube, with the addition of 1.5 µL of template DNA, one µL of each primer (10 pM/µL), and 16.5 µL of diethyl pyrocarbonate (DEPC)-treated water (20 uL total volume). PCR amplification was performed using

the following condition. Initial denaturation at 5 min at 95 °C, followed by 40 cycles of denaturation (30 s at 95 °C), annealing (30 s at 50 °C), and extension (30 s at 72 °C), with a final extension step (10 min at 72 °C). Additionally, to verify the quality of gDNA from non-target species, PCR was performed using universal COI primers (LCO1490/HCO2198; *Folmer et al., 1994*) under the same PCR conditions except for the annealing temperature of 45 °C.

We loaded a 1% agarose gel with two uL of PCR product stained with dye solution (BIONEER), a 100 bp DNA Ladder (Takara, Tokyo, Japan), and carried out an electrophoresis to confirm the amplification size.

## RESULTS

### Improvement in ascidian-specific primers

The primers AscCOI2 had the same length as AscCOI. Nevertheless, the target region, Tm (°C), and resulting amplicon size differed between primer sets. AscCOI2 was designed to amplify a 216 bp fragment of the COI gene. For the target region, the binding site of the AscCOI2_F was same as that of the AscCOI_F, whereas the binding site of the AscCOI2_R was shifted by 21 bp toward the 3′ end. The Tm values were 52.9 °C and 50.3 °C for AscCOI2_F and AscCOI2_R, respectively (Table 1). A GLMM analysis was performed to evaluate the difference between AscCOI and AscCOI2 primers. Results showed that AscCOI2, with odds ratio of 7.54, achieved significantly higher amplification success (estimate = 2.0206, SE = 0.2611, $p < 0.0001$) than AscCOI. The odds ratio of 7.54 indicates that AscCOI2 primers are approximately 7.5 times more likely to successfully amplify ascidian DNA than the AscCOI (Table S2).

Among the 1,674 bp length of the consensus sequence obtained from the alignment of 3,948 sequences, the AscCOI2 pair was located at 361–380 bp for forward and 553–572 bp for reverse. The proportion of the sequence was 100% at both AscCOI2_F and AscCOI2_R positions. However, the similarity index ranges differed for AscCOI2_F and AscCOI2_R, at 0.03–0.46 and 0.02–0.17, respectively (Fig. 1A). The maximum similarity index of AscCOI2_R (0.17) decreased compared to that of AscCOI_R (0.02–3.12). Primer binding analysis revealed that there was a clear difference between AscCOI and AscCOI2. *In silico* analysis showed that AscCOI_F and AscCOI2_F bind to 265 (97.07%) of the 273 species (100%) in the same way. However, AscCOI2_R showed significantly improved binding capacity by binding 226 species (82.78%), compared to 132 species (48.35%) bound by AscCOI. This improvement in the reverse direction makes the overall primer pair binding of AscCOI2 to be 225 species (82.42%) compared to 131 species (47.99%) for AscCOI. In the entire 3,948 sequence dataset (100%), AscCOI2 showed notably higher binding capacity than AscCOI. The AscCOI2_F was bound to 3,940 sequences (99.80%), whereas AscCOI was bound to 3,430 sequences (86.88%). The AscCOI2_R also demonstrated improved binding capacity, with 3,360 sequences (85.11%) compared to 2,977 sequences (75.41%) for AscCOI. The overall primer pair binding success of AscCOI2 (3,358 sequences, 85.06%) was significantly higher than that of AscCOI (2,465 sequences, 62.44%; Table 2 and Table S3). The results of the analysis of the non-target groups showed relatively low

**Table 2 Result of primer binding analysis.** Primer binding analysis comparing newly developed primers (AscCOI2) with previously developed primers (*Bae et al., 2023*). The analysis parameters allowed for a maximum of three bp mismatches throughout the primer sequence, and no mismatch was allowed two bp from the 3′ end.

| Primer | Target (total) | Forward (%) | Reverse (%) | Pair (%) |
|---|---|---|---|---|
| AscCOI | Species (273) | 265 (97.07) | 132 (48.35) | 131 (47.99) |
| | Sequence (3,948) | 3,430 (86.88) | 2,977 (75.41) | 2,465 (62.44) |
| AscCOI2 | Species (273) | 265 (97.07) | 226 (82.78) | 225 (82.42) |
| | Sequence (3,948) | 3,940 (99.80) | 3,360 (85.11) | 3,358 (85.06) |

potential binding capacity across the entire taxa (Table S4). Gymnolaemata had the highest binding capacity (11.06%, 24 out of 217 species) at the species level, whereas Anthozoa had the highest binding capacity (5.10%, 431 out of 8,449 sequences) at the sequence level. Other taxa had very low or no potential binding capacity with AscCOI2 (species level: 0% to 2.33%, sequence level 0% to 3.76%).

## Primer performance evaluation

Penalty score analysis revealed distinct differences between AscCOI and AscCOI2. AscCOI, tested only on ascidians, showed variable performance: the AscCOI_F demonstrated high efficiency with a low penalty score (median = 18.4), whereas the AscCOI_R showed low efficiency with a high penalty score (median = 553.6). Conversely, AscCOI2 showed significant improvement in ascidians, with both AscCOI2_F and AscCOI_R having low penalty scores (median = 18.45 and 120.4, respectively), which indicate enhanced binding efficiency. When AscCOI2 was tested on non-target groups, AscCOI2_F showed varying performance from Thecostraca and Chondrichthyes (median = 11.7), which had a lower value than Ascidiacea and Bivalvia (median = 282.9). However, AscCOI2_R consistently showed a higher penalty score in all non-target groups ranging from Chondrichthyes (median = 295.1) to Bivalvia (median = 787.8). Despite the low AscCOI2_F penalty score of Thecostraca, Echinodermata, and Chondrichthyes, all the non-target groups had AscCOI2_F and AscCOI_R penalty scores of below 150, owing to its high reverse primer score (Fig. 2).

The performance of primers was evaluated based on a threshold value of 150. For Ascidiacea, 3,934 (99.64%) sequences failed and 14 (0.35%) provided insufficient evaluation data when using the primer pair AscCOI. In contrast, AscCOI2 had 2,464 (62.41%) sequences that worked, 1,472 (37.28%) sequences that failed, and 12 (0.30%) sequences with missing data, indicating a relatively improved penalty score for AscCOI2. In addition, AscCOI2 was evaluated for the non-target groups to assess specificity. No sequence worked for all taxa in the non-target groups, and the sequences of Anthozoa (8,431 failure, 99.78%), Bivalvia (2,913 failure, 99.83%), Gymnolaemata (1,802 failure, 99.50%), Hydrozoa (4,453 failure, 99.80%), Porifera (6,468 failure, 98.07%), Thecostraca (14,029 failure, 99.33%), Echinodermata (19,888 failure, 99.91%), and Chondrichthyes (20,173 failure, 100%) all failed. The sequences with missing data varied from 0 (0%) to 127 (1.93%) throughout the non-target groups (Table 3).

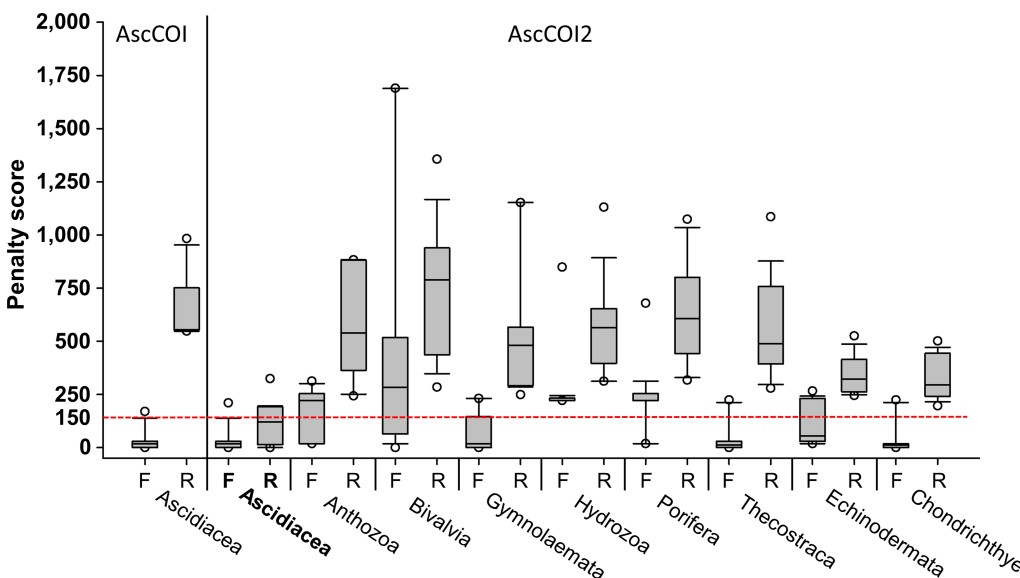

**Figure 2  Penalty scores for primers across target and non-target groups.** For AscCOI, only forward (F) and reverse (R) primer penalty scores for Ascidiacea are presented. For AscCOI2, penalty scores for both forward (F) and reverse (R) primers are displayed for Ascidiacea and six non-target groups (Anthozoa, Bivalvia, Gymnolaemata, Hydrozoa, Porifera, Thecostraca, Echinodermata, and Chondrichthyes). The box plot shows the median, quartile, and outliers for each primer direction. Horizontal red-dashed line represents the penalty score threshold of 150. Note that values below 150 are considered as "Working", whereas values above 150 are considered as "Fail".

**Table 3  Performance evaluation of the AscCOI and AscCOI2 for target and non-target group, with an established penalty score threshold of 150.** For each taxon, we provide the number of sequences analyzed, and the percentage of working, failure and missing data/gap s sequences. Working (%) indicates sequences below the threshold, failure (%) indicates sequences above the threshold, and missing data/gaps (%) indicates sequences for which the evaluation could not be completed. Bold values indicate results for the AscCOI2 on target group (Ascidiacea).

| Primer | Taxon | Total sequence (%) | Working (%) | Failure (%) | Missing data/gaps (%) |
|---|---|---|---|---|---|
| AscCOI | Ascidiacea | 3,948 (100) | 0 | 3,934 (99.65) | 14 (0.35) |
| AscCOI2 | **Ascidiacea** | **3,948 (100)** | **3,196 (80.95)** | **752 (19.05)** | **0 (0.00)** |
| | Anthozoa | 8,450 (100) | 0 | 8,431 (99.78) | 19 (0.22) |
| | Bivalvia | 2,918 (100) | 0 | 2,913 (99.83) | 5 (0.17) |
| | Gymnolaemata | 1,811 (100) | 0 | 1,802 (99.50) | 9 (0.50) |
| | Hydrozoa | 4,462 (100) | 0 | 4,453 (99.80) | 9 (0.20) |
| | Porifera | 6,595 (100) | 0 | 6,468 (98.07) | 127 (1.93) |
| | Thecostraca | 14,123 (100) | 0 | 14,029 (99.33) | 94 (0.67) |
| | Echinodermata | 19,905 (100) | 0 | 19,888 (99.91) | 17 (0.09) |
| | Chondrichthyes | 20,173 (100) | 0 | 20,173 (100) | 0 (0.00) |

The specificity of the AscCOI2 was validated *via in vitro* PCR tests using 22 genomic DNAs, comprising target groups (six) and non-target groups (16). PCR amplification was successfully performed on all six species of target groups, which were *Didemnum*

*vexillum*, *Ascidiella aspersa*, *Ciona robusta*, *Ciona savignyi*, *Styela plicata*, and *Herdmania momus*, with a clear single band of the expected size (216 bp). However, no amplification was observed in 16 species of non-target groups (Table S1; Fig. S1). The gDNA quality of the non-target group was confirmed using universal COI primers (LCO1490/HCO2198), which successfully amplified all samples except *Anthopleura fuscoviridis* (lane 20; Fig. S2). These results suggest that the AscCOI2 are specific to ascidian DNA and are unlikely to amplify DNA from other marine benthic invertebrate.

## Barcoding gap analysis

The results of barcoding gap analysis using the AscCOI2 showed that the intraspecific and interspecific variation had different patterns. Intraspecific genetic distances showed a clear unimodal distribution from 0% to 6.5%, with most values concentrated below 2% and the highest frequency occurring around 1%. In contrast, interspecific genetic distances were much wider, ranging from 8% to 91.2%, and showed a markedly different bimodal distribution. The distribution showed two distinct peaks. The smaller peak at about 11% appeared to represent differences among relatively close related species, whereas the more pronounced larger peak at about 35% indicated that the genetic differences among species were relatively larger. In addition, a less pronounced intermediate-sized distribution pattern was also observed from about 20% to 30%. The interspecific distance showed a long right tail that exceeded 60%, indicating that there was a significant genetic difference between some species pairs. This pattern was markedly different from the unimodal distribution of intraspecific distances, creating a clear barcode gap (1.5%) between the highest intraspecific (6.5%) and lowest interspecific (8%) genetic distances (Fig. 3).

## DISCUSSION

### Improvement in primer design and performance

This study was conducted to improve the insufficient species detection coverage of ascidian-specific primers in previous studies (*Bae, Kim & Yi, 2023*). Strategic modifications to the binding site and sequence were made, and verification using *in silico* and *in vitro* methods revealed that the redesigned primer pair had a significantly improved species detection ability compared to the previous primer pair. This demonstrates that our target modification successfully improved the detection coverage of species, which was a major limitation in the previous primers. Thus, these new primers are a more effective tool for comprehensive diversity assessment using mini-barcoding, especially when applied to metabarcoding (*Madduppa et al., 2021*; *Lavrador et al., 2024*).

The main improvement was in the strategic location of the reverse primers within the COI gene region. Similarity index analysis revealed that the AscCOI2 pair targets regions with high sequence coverage, with 100% of the sequences present at both binding sites, and produces an appropriate amplicon size (216 bp) that is suitable for metabarcoding applications. In the new position, the maximum similarity index of the reverse primer decreased from 3.12 to 0.17 but still provides sufficient binding specificity, as demonstrated by the relatively low penalty score and improved performance from the validation results. In particular, the similarity index at the two bases on 3′ ends of the primer, which is

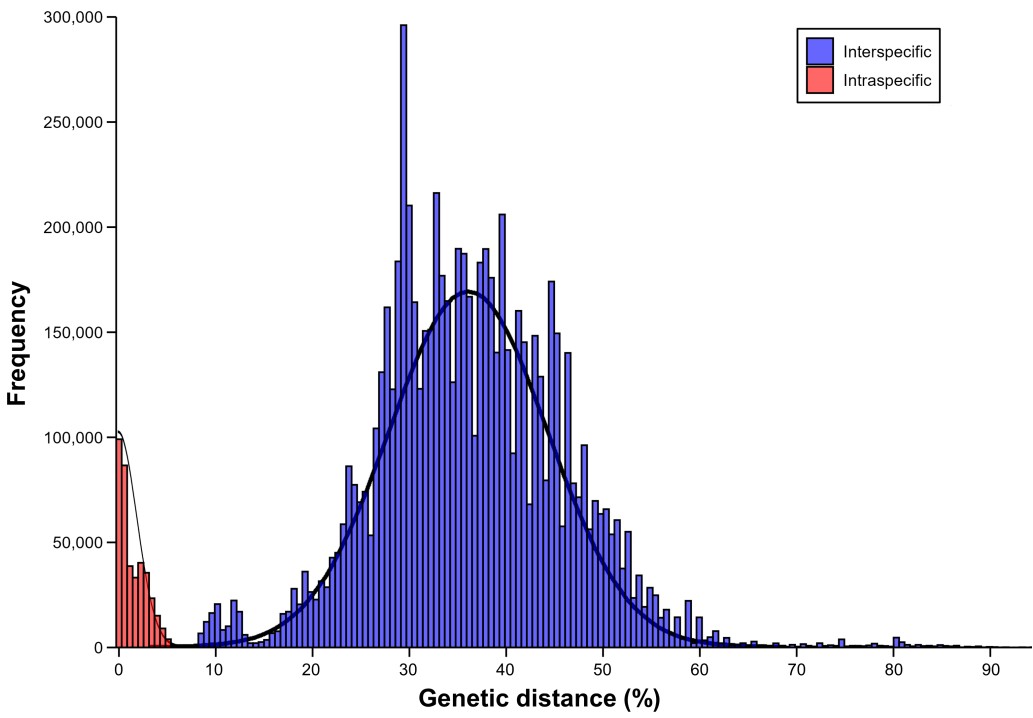

**Figure 3 Result of barcoding gap.** Frequency distribution of intraspecific (red) and interspecific (blue) genetic distances calculated using only ascidian dataset.

important for primer specificity (*Ye et al., 2012*), is between 0.11–0.23 for AscCOI and 0.11 for AscCOI2, even though no ambiguous base sequences were used in this region.

Primer binding analysis shows that AscCOI2 performs significantly better than AscCOI. The sharp increase in the binding success rate from 47.99% to 82.42% at the species level and from 62.44% to 85.06% at the sequence level indicates the effectiveness of our design strategy. This improvement is particularly noticeable in the reverse primer and believed to have had the greatest impact on the binding ability of the entire primer pair. The improved performance is due to the strategic selection of reverse primer binding sites based on sequence conservation throughout ascidians and matching at the 3′ end of the primer (*Ye et al., 2012*; *Freeland, 2017*). This significant improvement in primer binding efficiency is expected to improve the feasibility of ascidian diversity assessment.

## Validation of primer specificity and efficiency

Binding analysis of non-target groups demonstrated their potential to bind to Gymnolaemata and Anthozoa. This potential binding capacity may reduce specificity, which may be expected to improve under optimized PCR conditions (*Ficetola et al., 2010*) than *in silico* prediction. Furthermore, the comparatively low binding capacity of the non-target groups (up to 11.06% at the species level) compared to the binding capacity observed in ascidians (82.42%) indicates the specificity of AscCOI2 for ascidians.

Penalty score analysis further confirms the improvement in design. Based on the median, only the AscCOI2_F and AscCOI2_R fall below the threshold of 150, whereas

the non-target groups have relatively high score above 150. For echinoderms, *Hoareau & Boissin (2010)* demonstrated improved performance specificity of phylum-specific primers, with optimized primers clearly differentiating between target and non-target groups. In particular, the significantly higher penalty score in the reverse primer of the non-target group (ranging from 481.6 to 787.8) compared to that of our target group (∼120.4) indicates that AscCOI2 has a strong specificity for ascidians. This specificity is especially important for identifying target groups in marine environmental samples where multiple taxa coexist (*Govender et al., 2019*).

A relatively low penalty score was observed for AscCOI2_F in Thecostraca, Echinodermata, and Chondrichthyes, but was much higher for AscCOI_R, effectively preventing forward and reverse amplification from intersecting (*Vamos, Elbrecht & Leese, 2017*). Results of the primer performance evaluation showed that most of the taxa failed (98.07–99.83%) on the basis of the threshold value of 150, whereas most of the ascidians worked (80.95%), indicating that they were relatively successful. These results are similar to those of other metabarcoding studies that successfully evaluated primer performance at similar thresholds of 100–200 (*Bylemans et al., 2018*). Furthermore, the consistently high failure rate (over 98%) in all non-target groups, ranging from Porifera (98.07%) to Bivalvia (99.83%), corroborates the taxonomic specificity of AscCOI2.

Despite the high specificity of AscCOI2, the primer pair showed a failure of 19.05% for ascidian sequences. This failure is considerably lower than that of AscCOI (99.65%), indicating a significant improvement in performance. The remaining failures are likely attributable to ascidian species with higher genetic divergence at primer binding regions, consistent with findings in other taxonomic groups where primer-template mismatches affect amplification success (*Piñol et al., 2015*). Overall, the relatively high performance of AscCOI2 supports the fact that it provides a high working rate for comprehensive biodiversity assessment of ascidians while maintaining the high specificity required for metabarcoding applications.

Conventional PCR using AscCOI2, performed *in vitro*, amplified only ascidian DNA and produced clear single bands of the expected size (216 bp) but showed no amplification in any of the non-target species tested. This provides empirical support for our *in silico* predictions of primers specificity. This primer pair specificity (ability to exclude non-target species) is especially important for increasing the reliability of metabarcoding studies in complex marine environments with multiple taxonomic groups (*Collins et al., 2019*; *Kumar et al., 2022*). The consistent results between computational predictions of high penalty scores in non-target groups and experimental validation (selective amplification of only ascidian gDNA) suggest that AscCOI2 will likely maintain its high specificity even in environmental samples containing diverse marine invertebrate taxa. This dual validation approach of combining both *in silico* and *in vitro* analysis provides robust evidence for the utility of primers in metabarcoding, similar to successful strategies employed in other taxonomic groups (*Zhang, Zhao & Yao, 2020*).

### Barcoding capacity

The empirical threshold for species identification in commonly used COI genes is 1–3% (*Hebert et al., 2003a*; *Hebert, Ratnasingham & de, 2003b*; *Ratnasingham & Hebert, 2007*; *Zhang & Bu, 2022*). The higher threshold range (0−6.5%) obtained in this study is considered to result from our mini-barcode approach, which targets shorter COI fragments, and the greater genetic distance within ascidians compared to other benthic invertebrates. Moreover, the presence of a distinct barcode gap (from 6.5% to 8.0%) indicates that AscCOI2 effectively separates intraspecific and interspecific genetic distances in the ascidians. These results were achieved through the mini-barcode approach in ascidians, which amplifies short fragments (∼216 bp) with one pair of primer (*Meusnier et al., 2008*). Although mini barcodes have been effective in various taxa, our results demonstrate that this approach is particularly suitable for ascidian species identification and metabarcoding. Strategically designed short amplicon can provide better amplification success for DNA fragments (*Hajibabaei & McKenna, 2012*). However, specific primers can improve the recovery of particular taxonomic groups, which might be relevant from a commercial or ecological point of view, as well as being applied to other specific disciplines such as biodiversity assessment, detection of cryptic species, and metabarcoding for community evaluation and/or prey-detection studies (*Hajibabaei et al., 2006*; *Baumsteiger & Kerby, 2009*; *Saitoh, Uehara & Tega, 2009*; *Smith & Fisher, 2009*; *Rougerie et al., 2011*).

The distribution of interspecific genetic distances showed a bimodal pattern, which is different from the unimodal pattern in the previous barcoding gap analysis of AscCOI (*Bae, Kim & Yi, 2023*). The smaller peak (11.0%) represents recently diverged species pairs, whereas the larger peak (35.0%) represents deeper evolutionary divergences between species with more distant relationships (*Paz & Rinkevich, 2021*). A similar bimodal pattern has been observed in other marine organism (*Čandek & Kuntner, 2015*; *Zhang et al., 2023*; *Lutz et al., 2024*), and the potential intermediate elevations between the two distinct peaks might suggest additional hierarchical structure (*e.g.*, family, genus, and class) in genetic divergence (*e.g.*, 0.2, 0.25, and 0.3; *Kvist, 2016*); however, further investigation is needed to confirm this.

## CONCLUSIONS

In this study, we redesigned and validated the AscCOI2 pair, an improved pair of mini-barcode primers for molecular identification of ascidians. Through strategic primer design and comprehensive validation using both *in silico* and *in vitro* approaches, we significantly improved the success rate of amplification and maintained high taxonomic specificity (from 47.99% to 82.42% at the species level). The clear gap in the barcoding gap analysis confirmed the reliability of the primers for accurate species identification. These improvements suggest that the improved primer pairs are an effective tool that can be used for the assessment of biodiversity of ascidians based on eDNA metabarcoding. In addition, this effective tool can be used as a versatile molecular marker for a variety of applications, such as species identification and phylogenetic analysis.

### Funding
This study was supported by the grants of National Marine Biodiversity Institute of Korea (2025M00300). The funders had no role in study design, data collection and analysis, decision to publish, or preparation of the manuscript.

### Grant Disclosures
The following grant information was disclosed by the author:
National Marine Biodiversity Institute of Korea: 2025M00300.

### Competing Interests
The author declares that there are no competing interests.

### Author Contributions
- Seongjun Bae conceived and designed the experiments, performed the experiments, analyzed the data, prepared figures and/or tables, authored or reviewed drafts of the article, and approved the final draft.

### Data Availability
The raw measurements are available in the Supplementary Files.

### Supplemental Information
Supplemental information for this article can be found online at http://dx.doi.org/10.7717/peerj.19671#supplemental-information.

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
