# Peer review of "Validation of improved cytochrome c oxidase I (COI) primers for comprehensive biodiversity assessment of ascidians"

_PeerJ, doi:10.7717/peerj.19671_

## Round 0.1 · original submission · Major Revisions

Dear Authors, two reviewers read the manuscript and they both agree that the research is solid, well done and with enough data to support the results and the discussion. Nevertheless, they provided several useful comments that need to be addressed. In particular, they found several grammatical errors that can be fixed by improving English. Please take into account all the comments/suggestions to better improve the work.

**Language Note:** The Academic Editor has identified that the English language must be improved. PeerJ can provide language editing services - please contact us at [email protected] for pricing (be sure to provide your manuscript number and title). Alternatively, you should make your own arrangements to improve the language quality and provide details in your response letter. – PeerJ Staff

·

Basic reporting

The manuscript is well structured, with a solid background and relevant references (which might be slightly improved as indicated below). Figures are relevant and comply with PeerJ quality standards, and both tables and figure labels are described. Raw data (the new primer sequences) are provided as tables. I consider English should be improved, and I have tried to provide some clues to resolve this issue (most of my comments aim to improve it).

Experimental design

The experimental design is strong, including more than 40,000 cox1 sequences in the study to generate robust results. The research question is well defined, and relevant for those researchers on ascidian species, both on systematics or biomonitoring with eDNA metabarcoding, as many species are invasive. The investigation has performed with high technical and ethical standards. Methods are described, although I would recommend including some additional information to enable its replication.

Validity of the findings

The findings will be replicated on real research studies, and this will be used in future mini-barcode and metabarcoding studies. All data is provided (some points are raied in the general comments section), they are robust, sound, and controlled. Conclusions are well stated, linked to the original research question and limited to supporting results.

Additional comments

Bae, in the present study, redesign the primer set AscCOI, obtaining a new AscCOI2 version with significantly improved performance. To do so, they use large datasets containing ascidians (target dataset) and non-ascidian (non-target dataset) cytochrome oxidase 1 (COI) matrices. They use these datasets to in silico evaluate the similarity index of key regions for mini-barcoding on which redesign the primers. Following, they evaluate their performance through penalty score analysis, and perform a comparison among target and non-target taxa. They evaluate whether the region sequenced with the new primer set AscCOI2 present of a barcode gap between intraspecific and interspecific genetic distances, which might be useful for metabarcoding analyses. Finally, they synthetize and test them in vitro though conventional PCR on target and non-target species.
I consider this manuscript will be of great interest for researchers on ascidians, as well as for biologists and ecologists interested in biomonitoring invasive species, since invasive ascidians are a major threat to marine ecosystems. Overall, I consider that the work conducted by Bae is solid and might serve as a useful resource in the near future.

Although I really liked the manuscript, I have a few concerns that should be addressed. Most of them regard the English (which should not impact on the manuscript publication), and only a few regarding the materials used. Moreover, I have made some suggestions which I consider would improve the manuscript, although not necessary (marked with an OPTIONAL).

Across the manscuript
There is a lack of consistency between “primer set” and “primer pair”. Homogenize please to make it easier for the reader
Usually, units are separated from the numerical value by spaces. However, %, ºC and currency are exceptions, and are written together (e.g.: 9%, 16ºC, $1535). Please correct across the manuscript.
Instead of using the expressions “the forward/reverse primer of the AscCOI2”, I consider that using AscCOI_F and AscCOI_R helps the reader to follow easily the manuscript, as otherwise is too verbose. This should be corrected across the manuscript.
VERY OPTIONAL: Have you tried to in vitro amplify an ascidian species which failed when doing the in silico approach? I think it would be interesting to make sure that the software works correctly. In case you successfully amplify it, you can mention that the performance of the primer might be even better than obtained by in silico methods.

Abstract:
17: This sentence lacks relevance. I would do emphasis on the invasive nature of many ascidian species, and that they are difficult to identify. “they can be used in ascidians species, which are morphologically difficult to identify and suppose one of the most invasive taxa worldwide.”
18: remove “previously designed”
19: change improved by “new”
20: This paper does not conduct metabarcoding itself, although it proposes a nice set to do so in the future (yet it must be tested). Remove “through DNA metabarcoding”
23 “showed the improved efficiency of”
25: a score above 480
28: “of six ascidian species”
28: Say something about the failure of non-ascidian taxa as validation test.
31: The first part of the sentence sounds repetitive to the previous one. I would melt them and start the sentence with an: “Overall, this study demonstrate that the AscCOI2 primer set is an effective…”

Introduction:
Overall, I miss a small paragraph talking about what DNA barcoding is and its applications (species delimitation, mostly). Cox1 in animals, ITS in fungi, 16S bacteria, etc… You can go straight to animals, but please, include it to provide a more robust background. You talk about the barcode gap in Results and discussion, but you never talk about the cox1.
36: change “to effective” for “for effective”
38: “heavily relied”
42 “comprehensive approach that”
45: “ascidians, many of them with an invasive potential”
46: However, the effectiveness of these tools cannot be successfully used unless…”
48: “Biodiversity using eDNA techniques is particularly difficult to monitor in marine ecosystems, since DNA”
50: Remove HTS, as this is the only time it appears
53 “marine eDNA metabarcoding”
55: Rephrase for clarity. Do you want to say that many studies have been carried out in marine environments using eDNA, or that many studies have focused on developing new specific primers for each taxon?
57: Problem with reference manager. One of those should be separated outside of the Brackets to link with the remaining sentence.
59: This is the first time that the acronym COI appears. Please, define “cytochrome oxidase I”
61: Remove “the”
62: Rephrase. Not clear the meaning.
62-63: Split the sentence in two. I would Split it at (Bae et al., 2023ª). However, ….
64: “improve the ascidian-specific primer set to enlarge its spectrum of species detection”
69: Set the main target of the study: “generate a new ascidian-specific primer set (AscCOI2) by improving the previous set AscCOI (Bae…).
71-73: Establish how you Will get it (sub objectives). “ To ensure the quality of the new set, this study aimed to (1) verify through in silico analysis the primer-template …. And (2) confirm the performance of the new primer set by following in vitro analysis through conventional PCR. “
74: Results are pretty solid. You totally demonstrate the performance of your primers. I suggest be direct and use: “we demonstrate”
75: Change the order of mini-barcode and metabarcoding, as metabarcoding is one application of mini-barcoding (hierarchically speaking)

Material & Methods:
The work done in this paper is commendable, and the amount of data gathered is awesome. However, I noticed that you tested your primers in vitro with some taxa not tested in silico. Please, can you include these taxa in the in silico analysis? Moreover, although being invertebrates, ascidians are phylogenetically closer to vertebrates rather than invertebrates. I strongly encourage to in silico test fishes and echinoderms, as these are the closest taxa to ascidians (besides other biodiversity-poor taxa such as amphioxi, etc…). I would also like to see (to me it is not necessary as your results are extremely clear, although it is requested by the journal) a statistical analysis (such as GLMM) comparing the performance of both primers directions, to conclude the improvement is significantly better.
79: “For dataset construction”
82-84: Define how were ascidian cox1 sequences selected. Did you really only used complete mitochondrion genomes from GenBank? I did a fast search for cnidaria and I only retrieves ~2500 hits (see screenshot):

84-88: Table S1 is missing on the main manuscript. Can you provide the Genbank Accession code for each sample used in Table S1? It would be also nice if you provide a supplementary file with the ascidian alignment, although I will not insist on that (GitHub works really well to upload this kind of data ^^).
89: (VERY OPTIONAL) Among align algorithms, Clustal performs quite bad. Have you tried MAFFT or MUSCLE? Those perform overall much better, and perhaps your similarity index might improve.
91: “The forward primer (herein named as AscCOI2_F)”. Same for the reverse throughout the manuscript. Otherwise it is complex to follow when you talk about the F and the R , or the whole set.
91-93: Rephrase for clarity. Not clear what AscCOI and AScCOI2 are. Maybe the previous hint might help you to make the sentence easier to follow.
94-95: “similarity index analysis, calculated as …”
97: Can you define the parameters for the sliding window? Both window and step.
106 “Geneious Prime” for consistency.
107: Justify the maximum mismatch values you established. Maybe a reference?
109: Can you define the penalty settings?
112-114. I consider that this sentence match better after line 117 to justify your decision.
126-128. The PCR kit is one of those which are tubes with all reagents inside, right? If so, I would rephrase as: “Each PCR was carried out in an AccuPower PCR PreMix tube, by adding on them 1.5uL, …… (20 uL total volume)”
130-133: “ We loaded a 1% agarose gel with 2uL of PCR product stained with dye solution, a 100 bp DNA Ladder, and carried out an electrophoresis to confirm…”.
135: This paragraph can be embedded in the previous one, to make it more straightforward, as these computation analyses refer to lines 108-111. If so, you can make and in-silico section and an in vitro section.
138: Which function of the Biostrings package? Any remarkable parameter should be highlighted? The link is not necessary if properly cited in the references.
140: Which function of the PrimerMiner package? Parameters?
142: Which software was used to calculate the genetic distance with the K80 model? Which dataset did you use? Only ascidian sequences or all of them including non-target?
143: Function and parameters?
143-145: when writing the DNA barcoding paragraph within the introduction, this information would fit perfectly there.

Results:
The Barcoding gap analysis subsection is mostly written in present tense. Change it to the past tense in line with the rest of the Results
149-150: Rephrase to avoid redundancy / repetitiveness.
150 “pair had the same”
152-154: Reorder to fit the order established in line 151.
155: “the length of the consensus sequence, obtained from the alignment of 3948 sequences.”
157-159: Rephrase. I do not understand the meaning of what you intend to mention here.
161-162: Are you talking about the pair or only the forward? Not clear. Rephrase.
172: No binding against non-target taxa? I think it would be interesting to have those values, even if extremely-low, to acknowledge the limitations of the primers, if any., and confirm AscCOI2 is ascidian specific. If there is no binding, specify so as well, totally validating it.
176: “the ascidian dataset” / “ascidians”. Two times.
187-189: Remove this sentence as it says exactly the same as 185-187
204: “momus, with a”
205: Remove “was produced in the gel electrophoresis”
206-207: Three echinoderm species, four bryozoan species, and four arthropod species (Table S2”
211: I suggest transforming all values of the paragraph to %, as genetic distances are normally expressed as a decimal. This has an effect later on the discussion
214: “from 0% to 6.5%
215: 2% and 1%
216: “from 8% to 91%”
218: same
219: same
221: same
223-224: same
215: Use “in contrast” instead of “however”.
218 “differences among relatively close related species, whereas”
219: “indicated that the genetic differences among species were relatively larger, corresponding to non-related taxa
220: use “interspecific distance showed”

Discussion:
I miss a little bit of discussion about the nature of the primers, the amplicon size etc…. The Tms of the primers designed in this study are more similar between them than the Tms of AscCOI_F and AscCOI_R. For this reason, AscCOI2 might perform much better than the previous primer set AscCOI. Also, as AscCOI2 amplicons are 20 bp larger, can play a significant role if variable positions are find within the region, as it might allow a better taxonomic assignation and species identification.
232-233: I find this sentence repetitive. I would go for: “thus, these new primers will become a more effective tools for comprehensive diversity ssesment using mini-barcoding, specially when applied to metabarcoding (Maduppa…)”
237-238: What do you mean here by high-sequence content? I would rephrase for clarity
249-250: Here, or maybe on M&M / Results. Can you explain how did you carefully selected reverse binding sites? I think that my issue here is the word “careful”, which is a subjective term. If you remove this word, it might be OK as well. Although I would prefer a proper explanation in the M&M section.
254: “ascidian diversity assessment”
258: use “primer pair” in singular
259-260: Rephrase for clarity.
260: Remove brackets for Horeau & Boissin and keep it for (2010).
261: “improved performance”
262: “clearly differentiating”
263: add “in comparison to our target group (~120.4)”
269-270: “Primer performance evaluation showed that…”
270-271: Rephrase this part of the sentence, as I do not understand the meaning of it: “on the basis of the threshold value of 150, but ascidians worked, that is, it was relatively successful (80.95 %).”
273: “at similar thresholds of 100-200 (Bylemans…)”
274-275: “From porifera (98.07%) to Bivalve (99.83%)”
276: I miss here a little bit of comparison with Ascidians, as you only mention the failure of non-target groups but do not talk about the success (or failure ) of ascidians (19.05%)
277: This is a new paragraph. I would first state “Conventional PCR using AscCOI2”, performed in vitro, ….”
278: Remove “for all target species”, as it is repetitive with the previous “only ascidian DNA”.
280: “this primer pair specificity”. Otherwise can be understood as each one separated, being false for the AscCOI2_F.
283-284: As previously mentioned, you did not in silico and in vitro tested the same groups. You should include all taxa in vitro tested into the in silico analysis, in addition to fishes
284: Environmental samples also contain fishes. Please, include them into the in vitro analysis.
286-287: “This dual validation approach combining both in silico and in vitro analysis”
294: I have checked Bae et al 2023, as I was surprised by the values. Ascidians are characterized by an extremely rich intraspecific genetic diversity, given their large population sizes (even in invasive species which have suffered strong bottlenecks (10.1111/mec.17502) and, in some groups, the loss of DNA repair machinery (10.1007/978-3-030-23459-1_4). As suspected, I realized that those values indeed belong to 0-6.2% (instead of 0.000 – 0.062%), as genetic distance analyses provide percentages as decimals (per one instead as per cent). This high % value is in line with the genetic richness previously reported for this taxon. I would suggest remove “but in previous studies, the threshold in Ascidian was 0-0.062 (Bae et al., 2023a).”
296: Specify that the barcode gap goes from 6.5% to 8%. As that 1.5% is misleading (within the intraspecific genetic distance 0-6.5%)
298-299: In other taxa? Otherwise, I do not understand which is the discussion point here.
299-301: But also at higher taxonomic ranks, as the Leray primers. I would change “particular taxonomic groups” to “any taxonomic group”.
301-305: add here a plot twist, stating something like: However, specific primers can improve the recovery of particular taxonomic groups, which might be relevant from a commercial or ecological point of view, as well as being applied to other specific disciplines such as biodiversity assessment , detection of cryptic assessments and metabarcoding for community evaluation and/or prey-detection studies.”
308-310: Smaller peak of 11% may belong to genus and 35% to family/class levels? However, I see a peak at 10%, another at 20%, another at 25% and a huge one at 30%. These thresholds may correspond to sibling species, genus, family and class. I suggest elaborating on this observation. See QIIME (https://qiime2.org/) or USEARCH (https://www.drive5.com/usearch/)
311-313: I do not consider these are hidden phylogenetic lineages, but extremely distant lineages (e.g: hydrozoa against Ascidiacea).
316 “ an improved set of mini-barcode primers”
327-328: I guess this is hidden for reviewing purposes and potential conflict of interests, but be sure that the funding is here or they will complain! The sentence says grant”S” but only one code is provided.

Figures and Tables:
Figure 1: Instead of “green lines”, use “green histogram/areas”.
Figure 1: “The black and red dashed vertical lines indicate the positions of the AscCOI and AscCOI2 primer pairs, respectively”
Figure 2: You also compare AscCOI. Mention it in the legend, otherwise by only reading the caption it looks like you only include AscCOI2.
Figure 2: “Horizontal red dashed line represents the penalty score threshold of 150. Note that values below 150 are considered as “Working”, whereas values above 150 are considered as “Fail””
Figure 3: I t was not clear to me if the dataset used to obtain this plot was only ascidia or all samples. Specify it here as well. I would remove “representing DNA barcoding gap in ascidians”, as this kind of plots do not represent the barcode gap. The barcode gap is an interpretation of the data.
Table 2: Rephrase the second sentence, as it is difficult to follow right now. These are the parameters you defined to classify as good or bad? If so, mention that these values are parameters.
Table 3: First, I would place the column total sequences next to between Taxon and Working (%). Then, I would use the following first sentence: “Performance evaluation of the AscCOI and AscCOI2 primer pairs for target and non-target taxa, establishing a penalty score threshold of 150”, as ascidians are also marine invertebrates. Then I would continue with “For each taxon, we provide the number of sequences analyzed, and the percentage of working, failure and missing data/gappy sequences.”
Table S1: Include the GenBank accession codes of your samples.
Table SX: I miss a second table including all sequences used for non-target samples.

Reviewer 2 ·

Basic reporting

I tried to make suggestions for all spelling errors, but the article should be checked by a professional editor.

Experimental design

Methods

Line 123-124: In vitro validation of such a large dataset (3,948 target sequences and 38,359 non-target sequences) is cursory. In vitro validation is key to the utility of these primers for future researchers. Several of the non-target classes have no in vitro validation at all (i.e. Anthozoa, Bivalvia, Hydrozoa, Porifera). In vitro validation should include at minimum one species per genus for target sequences, and one species per family for non-target sequences.

Line 123-124: Controls with another COI primer should be conducted for every PCR on a non-target species, to test the quality of the DNA. If the goal is non-amplification, how does the researcher know whether non-amplification is the result of poor DNA quality, or the intentional primer design?

Validity of the findings

Please see Methods comments above.

Additional comments

Abstract
Line 17: “They can be used…” rather than “It can be used…”
Line 28: “…was observed between intra- and inter-species genetic distances in the barcoding gap analysis.”

Introduction
Line 37-41: However limited morphological taxonomy may be, it is still the only valid approach for describing a new species. If species with sequences on GenBank are not identified using morphological taxonomy or are incorrectly identified, or if species do not have sequences on GenBank, then barcoding is not useful. This should be acknowledged.
Line 39-41: “These traditional methods can potentially cause stress to organisms and disturb their habitats, and the effects vary greatly from taxon to taxon.” I would delete this sentence barcoding has the same downsides – you still need to collect the tissue.
For the limitations of morphological taxonomy in ascidians, I would look at the Introduction in Salonna et al. 2021 Scientific Reports, and Rocha et al. 2019 in Systematics and Biodiversity.
Line 51-52: Citation needed for “…the success of such studies critically depends on design and selection of primers.”
Line 55-58: Proofreading needed: missing subject?
Line 79: Sequence alignment should be provided.
Line 80: “Bivalvia” and “Gymnolaemata” – please correct spelling.
Line 85: How many species were included from each order of ascidians?
Line 86-87: “Bivalvia” and “Gymnolaemata” – please correct spelling.
Line 86: A list of the species should be provided in the Supplementary Information.
Line 90-91: “Primer binding regions were strategically placed to improve species detection coverage while maintaining ascidian-specificity.” This sentence should be expanded into a paragraph, to explain in more detail why the previous primers needed to be improved, and exactly how primers were placed to “improve species detection coverage” and “maintain ascidian-specificity”.
Line 90-91: Later in the Methods, the author states that the position of the Forward Primer did not change. This is not consistent with the sentence “Primer binding regions were strategically placed to improve species detection coverage while maintaining ascidian-specificity.” The placement of the original Forward and the new Forward is the same.
Line 90-91: What was the amplicon length of the previous primers, and what is the amplicon length of these improved primers? This information is found in Table 1, but should be stated in the text.
Line 90-91: When “primer binding regions were strategically placed”, that implies that the primer binding regions were changed between the primers published in 2023, and the primers in the current manuscript. By how much were the primer binding regions changed? This can be inferred from Table 1, but should be stated in the text.
Line 90-91: Please see Salonna et al. 2021 Figure 1 for an example of a figure showing the location of primers and the alignments associated with primer content. A similar figure would be very helpful for this manuscript.
Line 91-100: The similarity index analysis should be explained in more detail. What does “degree of degeneration” mean? What does “similarity” mean in this context – similarity between what and what? And how is “degree of degeneration” related to similarity?
Line 112-114: Please explain what median of the mid-range (100-200) refers to.
Line 123-124: In vitro validation of such a large dataset (3,948 target sequences and 38,359 non-target sequences) is cursory. In vitro validation is key to the utility of these primers for future researchers. Several of the non-target classes have no in vitro validation at all (i.e. Anthozoa, Bivalvia, Hydrozoa, Porifera). In vitro validation should include at minimum one species per genus for target sequences, and one species per family for non-target sequences.
Line 123-124: Controls with another COI primer should be conducted for every PCR on a non-target species, to test the quality of the DNA. If the goal is non-amplification, how does the researcher know whether non-amplification is the result of poor DNA quality, or the intentional primer design?
Table S1: Aplousobranchia is not a Family. M. argus is listed twice.
Table S2: Ascidiella aspersa is not in the Family Didemnidae, and Cionidae is not in the Order Aplousobranchia.
Table S2: Bugula californica – please fix mis-spelling.

Results
Line 156: Can you please explain this sentence, “The percentage of the sequence was 100% at both locations…”
Line 157: Can you please explain this sentence “but the similarity index values differed, with 0.03 to 0.46 for forward and 0.02 to 0.17 for reverse (Figure 1).” Differed between AscCOI and AscCOI2?
Line 176: “primers in ascidians?
Line 176: “tested only on ascidians..”
Line 183: “Bivalvia”
Line 185: “Bivalvia”
Line 190: “Ascidiacea”
Line 194: “…improved penalty score for AscCOI2”
Line 208: “ascidian”
Line 212: “The results of barcoding gap analysis using the AscCOI2 primers…”
Line 228: “ascidian-specific”

Discussion
Line 237-238: Please explain “high sequence content”
Line 237-238: How is optimal defined here?
Line 245: “significantly better than AscCOI”
Line 250: “ascidians”
Line 254: “ascidians”
Line 264: “ascidians”
Line 270: “taxa”
Line 271: What does “it” refer to here?
Line 294: “ascidians”
Line 294: The author(s) should cite ascidian barcoding studies other than their own.

---

## Round 0.2 · Minor Revisions

Dear Authors,

The reviewers have read and commented on the new version of the manuscript. They agree that you did a good job in addressing all their previous comments. Here you will find an annotated PDF with some small suggestions. Moreover, one of the reviewers recommends some (OPTIONAL) analyses to add statistical support, since your results are already quite solid, and to do some grammatical changes which would help to improve the readability of the manuscript. So my decision is minor revision.

·

Basic reporting

It mostly uses clear and unambiguous English. Background is sufficient. The structure of the article is fine and provides proper figures and tables. Manuscript self-contained.

Experimental design

It aligns with the scope. The hypothesis is well defined, and the objectives are clearly defined. It follows a rigourous and extensive technical and ethical investigation. MEthods are properly described.

Validity of the findings

The findings are extensively validated. The results are robust, statistically sound, & controlled. Conclusions are well stated, linked to original research question & limited to supporting results.

Additional comments

Bae did an excellent work between manuscript versions. The current version of the manuscript is written in a much clearer way, provides a more meaningful background, and carries out additional analyses to add statistical support to their findings. They have carefully addressed all my comments, and provided a rational argument when rejecting any of them, on which I agree.
Please, find below a few minor comments. As in the previous round of revisions, I recommend some (OPTIONAL) analyses to add statistical support, since your results are already quite solid. Moreover, I suggest some grammatical changes which I consider would help to improve the readability of the manuscript.
I consider that the present paper is close to have a perfect shape for its publication, and I really look forward to see it available to the world.
Cheers and thank you

Reviewer 2 ·

Basic reporting

I appreciate the author's efforts to address all of my comments. I am willing to accept this revised manuscript in its current form.

Experimental design

No comment.

Validity of the findings

No comment.

Additional comments

No comment.

---

## Round 0.3 · accepted · Accept

Dear Authors, I am happy to inform you that your paper has been accepted for publication in PeerJ.

·

Basic reporting

The english is clear and professional throughout the manuscript. Literature is well references, and the structure is perfectly set. All necessary data is shared. The article is self- contained with relevant results to answers its hypothesis

Experimental design

The research is aligned with the journal. The research question is well defined as a knowledge gap. Investigation follows high technical and ethical standwards. MEthods are described in detail for their replication

Validity of the findings

All data has been provided, it is robust, sound, and controlled. Conclusions are well stated, linked to the original question and limited to supporting results

Additional comments

Bae did an excellent job adressing all of my concerns. I consider that the manuscript is ready for its publication. I want to congratulate Dr. Bae for his "solo" work, and look forward to see the applicability of these new primers in future studies.